# Diagnostic Concordance between Optical Coherence Tomography and Histological Investigations for Immune-Mediated Desquamative Gingivitis: Observational Study

**DOI:** 10.3390/ijerph18179095

**Published:** 2021-08-28

**Authors:** Vera Panzarella, Alessia Bartolone, Rita Coniglio, Vito Rodolico, Laura Maniscalco, Giorgia Capocasale, Martina Iurato Carbone, Giuseppina Campisi

**Affiliations:** 1Department of Surgical, Oncological and Oral Sciences (Di.Chir.On.S), University of Palermo, 90127 Palermo, Italy; alessia.bartolone@community.unipa.it (A.B.); martina.iuratocarbone@unipa.it (M.I.C.); giuseppina.campisi@unipa.it (G.C.); 2Sector of Oral Medicine, Azienda Ospedaliera Universitaria Policlinico (A.O.U.P.) “Paolo Giaccone” of Palermo, 90127 Palermo, Italy; ritaconiglio@gmail.com; 3Department ProMISE, University of Palermo, 90127 Palermo, Italy; vito.rodolico@unipa.it; 4Department of Biomedicine, Neuroscience and Advanced Diagnostics, University of Palermo, 90127 Palermo, Italy; maniscalco.laura92@gmail.com; 5Department of Surgical Sciences, Paediatrics and Gynaecology, University of Verona, Policlinico “G. B. Rossi” of Verona, 37134 Verona, Italy; capocasalegiorgia@gmail.com

**Keywords:** tomography, optical coherence, desquamative gingivitis, immune-mediated diseases, diagnosis

## Abstract

Desquamative gingivitis (DG) denotes a heterogeneous immune-mediated disease for which early diagnosis represents a great challenge. The main aim of this study is to validate diagnostic concordance between specific Optical Coherence Tomography (OTC) patterns for DG related to oral Lichen Planus (OLP), Pemphigus Vulgaris (PV), and Mucous Membrane Pemphigoid (MMP) and definitive histological diagnosis. Forty-three patients with suspected immune-mediated DGs, were progressively recruited. Before biopsy, an OCT preliminary evaluation was performed using specific pre-determined OCT diagnostic patterns (i.e., morphology and localization of blisters, status of the basal membrane, epithelial thickness, presence/absence of acantholytic cells into blister and/or inflammatory infiltrate) related to OLP, PV and MMP. After histological confirmation, OCT and histological diagnoses were compared. Using pre-determined patterns, OCT diagnoses of DGs were: 22 (51%) OLP, of which 11 (26%) were with the bullous variant, 4 (9%) PV and 6 (14%) MMP. The same diagnoses were found by histological investigations (with the main OCT discriminatory potential for the bullous variant of OLP). The concordance between the two diagnostic methods was confirmed by the Fisher exact test (*p*-value < 0.01). These specific OCT patterns show a diagnostic reliability in 100% of the cases investigated, suggesting their accuracy to support the complex diagnosis and management of immune-mediated DGs.

## 1. Introduction

Desquamative gingivitis (DG) is a clinical definition to describe an oral condition characterized by erythema, epithelial desquamation, atrophy, ulcerations and/or presence of vesicle-bullous lesions in the gingival mucosa [1]; it denotes a clinical sign of a very large spectrum of diseases with different pathogeneses. Among these, Pemphigus Vulgaris (PV), Mucous Membrane Pemphigoid (MMP) and Oral Lichen Planus (OLP), represent about 80% of cases of DG [1,2,3].

These immune-mediated diseases could be characterized by muco-cutaneous involvement and chronic course [4]. However, almost one third of the patients present primarily only gingival involvement in the form of DG which remains under-diagnosed for a longer time [4]. The main cause of this ambiguity is not known, but it could be attributed to the less severe nature and to the chronic course of immune-mediated DGs. Moreover, a late diagnosis is often related to the non-specific features of the disease, clinically comparable with the large number of conditions that present themselves as gingival inflammation, especially those which are plaque related [5]. Because of the delayed diagnosis and inappropriate treatment, immune-mediated DGs lead to disease progression with multi-organ involvement, increased treatment burden and cost, prolonged patient suffering and impaired quality of life [4].

Consequently, efforts to improve the early and correct detection of these immune-mediated DGs are mandatory [6]. To date, the diagnostic algorithm for erosive/vesicular-bullous DG immune-mediated diseases includes the combination of clinical/anamnestic data (e.g., clinical onset, extra-oral involvement, Nikolsky’s sign), immuno-serological tests (i.e., indirect immunofluorescence, enzyme-linked immunosorbent assay) and histological examinations [6,7] with direct immunofluorescence (DIF) [2,4,8]. However, DIF is an expensive technique, available only in advanced research laboratories [9]; also, the practice of biopsy in gingiva affected by DG is difficult because the tissues are fragile and thin, so difficult to manipulate [7].

In this context, a real-time evaluation in vivo of the suspected gingiva could be strongly advantageous to guide a more accurate diagnosis approach, from the choice of the most appropriate site for the diagnostic biopsy, to the facilitated chronic, non-invasive, monitoring of the patient with confirmed immune-mediated DGs [10]. 

New optical imaging technologies can provide detailed images of tissue architecture; among these, Optical Coherence Tomography (OCT) technology is playing an increasingly important role in several areas of medicine (e.g., ophthalmology and dermatology). This tool provides high-resolution, micron-scale tomographic images of the micro-structural architecture of different tissue [11]. The technique could be an excellent non-invasive support in oral medicine practice, especially for the diagnosis and management of patients with oral chronic diseases, such us DG. OCT allows direct visualization of the lesions related to DG and thanks to its non-invasive nature, it easily consents the examination of multiple lesions in the same patient and in the same session; thus, increasing the probability of obtaining more meaningful discriminative data useful for the subsequent diagnostic process [12,13,14,15,16,17].

A very recent systematic review of the literature, investigating the feasibility of OCT for the preliminary evaluation of DG, allowed to identify potential diagnostic OCT patterns for immune-mediated DGs (i.e., MMP and PV) [18].

However, to date, no comparative evaluation has been conducted between these patterns and histological investigation, which represents the gold standard (together with DIF) for the definitive diagnosis of immune-mediated DG; consequently, there is no validation of their diagnostic accuracy.

In this scenario, the main aim of our study is to evaluate the diagnostic concordance between specific diagnostic OCT patterns of DGs related to OLP, PV and MMP, and a histological confirmatory examination in a cohort of patients with clinical suspected immune-mediated DGs.

## 2. Materials and Methods

This study conformed to the ethical guidelines of the 1964 Declaration of Helsinki and its later amendments or comparable ethical standards. It was also approved by the Institutional Review Board of the ‘*Paolo Giaccone*’ Policlinico University Hospital (A.O.U.P.) in Palermo (Italy) (approval number 11/2016). 

### 2.1. Entry Criteria

Patient recruitment commenced on 1 May 2019 and finished on 31 December 2021. All participants were consecutively recruited from the Sector of Oral Medicine, A.O.U.P. “P. Giaccone” of Palermo (Italy). 

The eligibility criteria were: (i)Age ≥ 18 years;(ii)Ability to provide informed consent;(iii)Suspected DG related to MMP, PV or OLP, with major involvement of the vestibular gingival masticatory mucosa of the anterior sextants of the oral cavity (i.e., III and V sextants and the most medial portions of I, II, IV and VI sextants) *.** This criterion was imposed by the choice of OCT probe used in this study, characterized by a structure and size that did not allow the management of lesions in the posterior regions of the oral cavity (details later)*.

### 2.2. Data Collection and Clinical, OCT and Histological Examinations

Patients with entry criteria were progressively recruited and interviewed, using a structured, pre-tested baseline questionnaire. During the interview, variables, including socio-demographic data, medical history (i.e., presence/absence of immune-mediated diseases and any related treatments), were recorded. To assess health-related variables, the patients were interviewed as to their current and lifetime smoking history and alcohol consumption. Regarding tobacco use, patients were classified as never, current and former smokers (if they had quit smoking at least a year prior to the study). Smokers were classified into categories according to their cumulative tobacco consumption: (a) 0 (non-smokers), (b) <25 pack per year (light smokers) and (c) ≥25 pack per year (moderate/heavy smokers). Alcohol consumption was defined in terms of drink units (DU) per year: (a) non-drinkers (who had never consumed alcohol or who had less than one drink per year); moderate drinkers (<16 DU per year), and heavy drinkers (≥16 DU per year).

All patients signed written informed consent prior to the following diagnostic procedures.

*Clinical examination.* Inspection and palpation of oral cavity, with particular attention to the gingival mucosa, were conducted for all participants by a single oral medicine specialist (V.P.) according to the method proposed by the World Health Organization. To discriminate cases of DG clinically associated with vesicular–bullous diseases, any positivity to Nikolsky’s sign was recorded. Moreover, an inspection of the extra-oral muco-cutaneous sites was performed to detect any lesions potentially attributable to Lichen Planus versus PV/MMP. Digital photos were archived for each patient to record the clinical site of OCT and histological investigations. For this purpose, a Nikon D7200 Camera with Nikon AF-S DX 105 mm F2.8G, was used.

*OCT evaluation.* The evaluation of patients with suspected DGs related to MMP, PV or OLP was performed with OCT instrument (SS-OCT VivoSight^®^, Michelson Diagnostics Ltd., version 2.0, Orpington, Kent, UK). The light source had the following parameters: wavelength of 1305 ± 15 nm, axial optical resolution of <10 μm and lateral optical resolution of <10 μm, with a maximum image width of 6 mm × 6 mm and a focal depth of ≈2 mm. The probe used, measuring approximately 23 × 12 cm, consisted of a handpiece to allow proper handling and a flexible extension crossed by the optical fiber. No site preparation was required for evaluation. A plastic spacer of about 5 mm was applied to the probe to calibrate and facilitate its positioning on the mucosal surface and to set the correct scanning distance. The spacer can also be rotated to set the most appropriate orientation for the area to be scanned (Figure 1).

A scan of 6 mm in width and 2 mm in depth was performed, with a distance between frames of 0.1 mm, for a total of 60 frames for each scan, automatically stored in a specific database. 

From these, the best frames were finally selected for the OCT diagnoses, based on pre-determined patterns considered mainly discriminative for DGs related to OLP (erosive and bullous), PV and MMP [15,18], and after comparison with OCT archive image of gingival healthy mucosa (Figure 2). These patterns were summarized below.


*OCT pattern for erosive OLP:*

*Normal stratified epithelial layer and epithelial thickness;*

*Normal basal membrane and lamina propria;*

*Presence of intense inflammatory infiltrate.*




*OCT pattern for bullous OLP:*

*Presence of multilocular subepithelial blister;*

*Normal stratified epithelial layer and epithelial thickness;*

*Normal basal membrane and lamina propria;*

*Presence of intense inflammatory infiltrate.*




*OCT pattern for PV:*

*Presence of unilocular intraepithelial blister;*

*Reduced stratified epithelial layer and epithelial thickness;*

*Normal basal membrane and lamina propria;*

*Presence of acantholytic cells into the blister.*




*OCT pattern for MMP:*

*Presence of multilocular subepithelial blister;*

*Normal stratified epithelial layer and epithelial thickness;*

*Altered/indistinguishable basal membrane and lamina propria;*

*Presence of inflammatory infiltrate.*



*Histological examination.* The diagnostic protocol was completed with the biopsy and histopathological examination for all patients. After local anesthesia, an incisional biopsy was performed, in the same site of more representative OCT evaluation, using a 6 mm diameter punch (to ensure correspondence with the OCT scan performed at 6 mm width and 2 mm depth). Specimens were processed routinely in 10% formalin and embedded in paraffin. Formalin-fixed, paraffin-embedded (FFPE) sections of 5 µm were stained with routine haematoxylin and eosin and examined to perform histological diagnosis. The pathologist (V.R.) examined the images independently and in blindness from OCT results. OCT and histological diagnoses were ultimately compared. 

### 2.3. Statistical Analysis

Categorical variables were summarized as counts and percentages, discrete and continuous variables such as mean, standard deviation (SD) and range. The sample size calculation formula was applied to evaluate the proportion that required confidence level, error margin, and assumed percentage of correctly detected lesions as preliminary parameters:$$n=z_{\alpha /2}^{2}\frac{\hat{\pi }\left(1-\hat{\pi }\right)}{\delta^{2}}$$

With a confidence level of 95% and a 10% error margin, we estimated that *n* = 35 lesions were sufficient to estimate a percentage of 90% of correctly detected lesions by the two methods (OCT vs. histological evaluations). The Fisher exact test was used to assess the concordance between the two diagnostic tests. Statistical analysis was performed with the R software (version 4.0.2).

## 3. Results

Forty-three patients with DG potentially related to immune-mediated OLP, MMP and PV were consecutively enrolled. The patients presented the main involvement of the II (27/43, 63%) and of both the II and the V sextants of the oral cavity (8/43, 19%). All demographic and anamnestic data were collated in Table 1. This mostly related to females (37/43, 86%), with a mean age of 66.5 ± 14.1 years (range 30–90 years). No patients had the medical history of immune-mediated diseases. A total of 18 patients were moderate–heavy smokers (42%) and 5 patients were moderate drinkers (12%).

Based on the intra- and extra-oral clinical examinations, the following *clinically based diagnostic hypotheses* were taken:♦N° 19 cases (19/43, 44%) of vesicular–bullous DGs by OLP (10/19, 53%), PV (4/19, 21%) and MMP (5/19, 26%) with positive Nikolsky’s sign;♦N° 24 cases (24/43, 56%) of erosive DGs by OLP with negative Nikolsky’s sign.

The OCT examination, using specific selected patterns, led to the formulation of the following *OCT diagnoses* of DGs:♦N° 33 cases (33/43, 77%) of OLP, with sub-specific OCT diagnosis of the vesicular–bullous variant in n° 11/33 (33%) cases;♦N° 4 cases (4/43, 9%) of PV;♦N° 6 cases (6/43, 14%) of MMP.

The biopsy and histological examinations led to the following *confirmatory histopathological diagnoses:*♦N° 33 cases (77%) of DGs with a diagnosis of OLP (with identification of the bullous variant in only 1/ 33 cases);♦N° 4 cases (9%) of DGs with a diagnosis of PV;♦N° 6 cases (14%) of DGs with a diagnosis of MMP.

Clinical hypotheses, OCT and histopathological diagnoses are compared in Table 2.

Of 43 lesions, 4 (9%) were detected as PV, 6 (14%) as MMP and 33 (77%) as OLP, by both the OCT and histological examinations. The concordance between the two diagnostic methods was statistically significant (*p*-value of the fisher exact test <0.01). Noteworthy is that, of the 33 OLP lesions, OCT managed to identify 11 cases (33%) as bullous (versus only one intercepted by biopsy).

Figure 3 shows the comparison models performed for diagnostic concordance between OCT and histological images, for erosive/bullous OLP, VP and MMP. 

## 4. Discussion

The diagnostic process of immune-mediated DG was conducted through a combination of clinical report (e.g., clinical onset, extra-oral involvement, anamnesis, Nikolsky’s sign), laboratory data (i.e., immuno-fluorescence and immunoenzymatic tests) and histological examination [6,7,19,20]. 

Despite the specificity of Nikolsky’s sign was high in distinguishing between vesicular and non-vesicular bullous diseases, the technique was not always easy to perform [12], opening to a greater confusion in the preliminary clinical assessment [21]. The biopsy for histopathology and direct immunofluorescence (DIF) microscopy (considered the gold standard approaches) were not without complications [2,22]. The biopsy in the gingiva affected by DG was difficult because the tissue was fragile and thin and, therefore, arduous to manipulate [7], especially in the case of vesicular–bullous pathology in which, for a correct histological examination, the sample inclusion of the intact bulla with epithelium and connective tissues is necessary [7,9]. 

To overcome these complexities, the effectiveness of the OCT technique was tested, and some studies supported the OCT ability to preliminarily discriminate in vivo DGs [18]. The application of this device on gingival mucosa was facilitated by the non-elasticity of the tissue, reducing one of the main problems related to use of OCT in the oral cavity, and typically represented by altered optical properties induced by the mechanical compression of the oral soft tissue [14]. 

To date, the different OCT diagnostic patterns reported in the literature for immune-mediated DGs are based on information about the epithelium layers [12,15], basement membrane, lamina propria [13,15] and other intra/sub epithelial constituents (e.g., inflammatory, acantholytic cells) [23]. However, these patterns are not definitive and are not supported by studies with a confirmatory histological diagnosis.

In this context, the aim of this study was to test the diagnostic potential of specific OCT patterns of GDs potentially related to OLP, MMP and PV.

The epithelial layer, the basement membrane, and the lamina propria were the three morpho-graphic features used as reference in OCT identification of DGs. Specifically, the integrity and the thickness of the gingival epithelial layer were modified by both the presence of sub-epithelial or intra-epithelial blisters, respectively, in cases of bullous OLP/MMP or of PV, and by the presence of erosion/atrophy, reducing the epithelial lumen, in cases of atrophic–erosive OLP [15]. Basement membrane and lamina propria, always visible in a healthy gingival mucosa, in cases of DGs by MMP and OLP, were altered because of the presence of an inflammatory infiltrate [15,23].

Moreover, some other OCT determinants of vesicular–bullous DGs, such as fibrin residues and acantholytic cells into the blisters, respectively, in cases of MMP and PV, could guide a preliminary OCT assessment [23]. 

Based on these evidences, we performed specific pre-determined OCT patterns of DGs associated with erosive and bullous OLP, PV and MMP and tested their diagnostic concordance with histological definitive diagnoses in a population of 43 patients with a clinical suspicion of immune-mediated DGs.

The main finding of this observational study was the concordance between the OCT diagnosis and histopathological confirmatory analysis in 100% of the cases evaluated. 

Specifically, the OCT patterns allowed to discriminate all vesicular–bullous variants of the selected DGs pictured, thanks to the ability to detect the presence of an intra- and sub-epithelial blister in a higher and/or equivalent percentage of cases than the histopathological investigation (11/1 of bullous OLP, 4/4 of PV and 6/6 of MMP). In our case series, Nikolsky’s sign was positive in 10 out of 11 OCT cases of bullous OLP and in 5 out of 6 OCT cases of MMP, confirming the diagnostic potential of the method also with respect to the clinical preliminary evaluation. 

It is important to emphasize that these OCT discriminative advantages in vesicular–bullous lesions also guaranteed their evaluation without causing iatrogenic tissue damage, strongly associated with biopsy maneuvers, with the consequent alteration of the lesioned site and altered microscopic interpretation. 

The ease of execution, the non-invasiveness, the depth of penetration (up to 2 mm) and the accuracy of the OCT technique easily allowed the examination of multiple lesions in the same patient and in the same session; thus, increasing the probability of obtaining more preliminary significant data and guiding the execution of other confirmatory diagnostic investigations (i.e., biopsy/histopathological examination) [14,16,17].

The result of this study could have important repercussions not only in the diagnostic phase, but also in the management of the patient with immune-mediated chronic DGs, due the OCT’s ability to store and compare images at any time, allowing to monitor treatment and to identify disease relapse before clinical manifestations [24]. 

The OCT investigation of suspected immune-mediated DGs, conducted with a selection of pre-determined OCT patterns, would ensure:♦An additional diagnosis to the confirmatory histological one;♦A more appropriate identification of vesicular–bullous lesions and their site-specificity, both in the case of pure bullous pathology (i.e., MMP and PV) and in the case of bullous OLP, with or without positivity of Nikolsky’s sign;♦The identification of the most representative biopsy site of pathology.

The study, with its pioneering methodological characteristics, presented only the limitation of the relatively small sample size, which, however, is a probable reflection of the rarity of these diseases [4]. In the future, it will be necessary to validate these OCT discriminative patterns on a larger sample size to confirm the results obtained; this would open new possible diagnostic and management strategies for patients with immune-mediated DGs.

## 5. Conclusions

This study has allowed to validate a discriminative system of DGs, supported by OCT and able to distinguish in vivo, in real time and in a non-invasive way, the epithelial and subepithelial layer of the gingival tissue, guiding physicians in the differential diagnosis and in the choice of the biopsy site. 

In addition, in view of the endorsed patterns and the advantages hypothetically related to their use, it could be possible to create diagnostic supports, based on OCT computer analysis systems, able to process and compare pathological pictures of DG overcoming the limitations related to the operator and enhancing the diagnosis.

## Figures and Tables

**Figure 1 ijerph-18-09095-f001:**
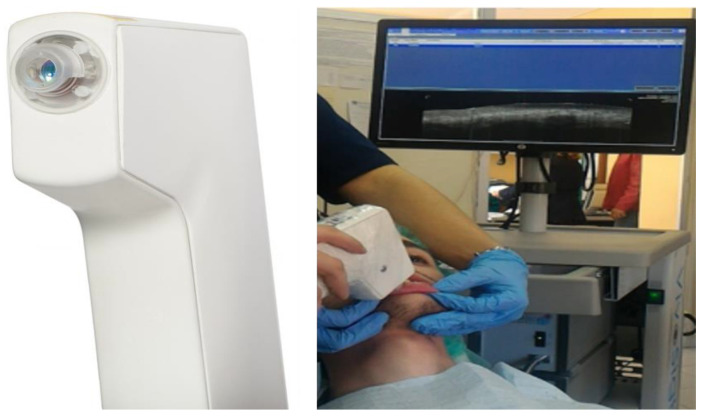
VivoSight^®^ OCT laser scanning handpiece (Michelson Diagnostics Ltd., Kent, UK), used for the study.

**Figure 2 ijerph-18-09095-f002:**
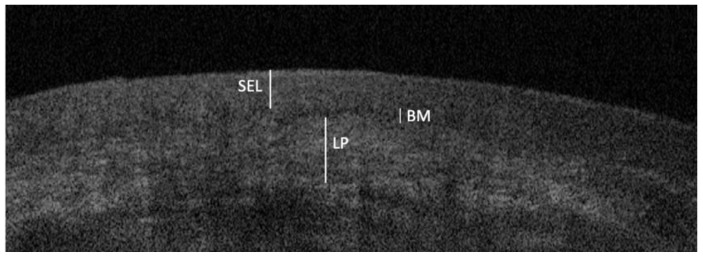
OCT image of gingival healthy oral mucosa: *Stratified Epithelial Layer* (SEL), characterized by a darker appearance due to the lower optical density, refractivity, and dispersion properties; *Basal Membrane* (BM), detectable by a low signal intensity, represents the transition zone between the epithelial layer and the underlying lamina propria; *Lamina Propria* (LP), characterized by a higher signal intensity and a more reflective white zone.

**Figure 3 ijerph-18-09095-f003:**
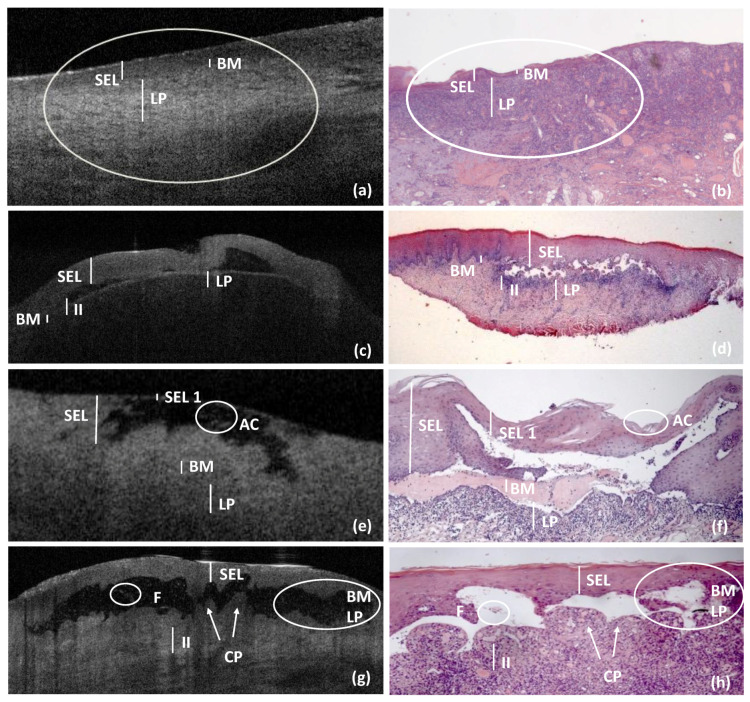
OCT and histological comparison models for DGs, by erosive and bullous OLP, PV and MMP. *DGs comparison models for erosive OLP* ((**a**) OCT section and (**b**) histological images (hematoxylin–eosin; original magnification 25×)): SEL present normal thickness, BM and LP appeared as normal, but both were difficult to detect due increased cellularity related to the inflammatory infiltrate, showing as punctiform structures into the underlying connective tissue (indicated with a circle). *DGs comparison models for bullous OLP*: ((**c**) OCT section and (**d**) histological images (hematoxylin–eosin; original magnification 10×)): multi-locular sub-epithelial blister, with inflammatory cells around and in the lumen of the blister, was present; it was evident the sub-epithelial detachment with a normal SEL and an LP visible as a linear band at the bottom of the blister associated with underlying increased inflammatory infiltrate (II). *DGs comparison model for PV*: ((**e**) OCT section and (**f**) histological images (hematoxylin–eosin; original magnification 10×)): unilocular intra-epithelial blister, with acantholytic cells (AC) inside, was present; SEL was strongly reduced. The roof of the blister was formed by the rest of the epithelium, which was intact with some occasional inflammatory cells; BM and LP were preserved and normally represented. *DGs comparison models for MMP*: ((**g**) OCT section and (**h**) histological images (hematoxylin–eosin; original magnification 10×)): multi-locular sub-epithelial blister, with fibrin residues (F) in the lumen and in the floor of the blister, was present; SEL was normal, and the connective papillae (CP) were preserved and identified the epithelial detachment. Due the inflammatory infiltrate (II) around the blister, it was not possible to identify BM and LP (indicated with a circle). *OLP—Oral Lichen Planus; PV—Pemphigus Vulgaris; MMP—Mucous Membrane Pemphigoid; SEL—Stratified Epithelial Layer; BM—Basement Membrane; LP—Lamina Propria; II—Inflammatory Infiltrate; CP—Connective Papillae; F—Fibrin residues; AC—Acantholytic Cells*.

**Table 1 ijerph-18-09095-t001:** Information regarding sex, medical history of immune-mediated diseases, tobacco (pack/year) and alcohol consumption (drink units—DU/year).

	DG (No. 43)	
	N	(%)
**Sex**		
Female	37	(86)
Male	6	(14)
**Medical history of immune-mediated diseases**		
Negative	43	(100)
Positive	0	(0)
**Tobacco consumption (pack/years)**		
Non-smokers (0)	21	(49)
Light smokers (<25)	4	(9)
Moderate/heavy smokers (>25)	18	(42)
**Alcohol consumption (DU/years)**		
Non-drinkers (0)	38	(88)
Moderate drinkers (<16)	5	(12)
Heavy drinkers (>16)	0	(0)

**Table 2 ijerph-18-09095-t002:** Comparison between clinical hypotheses, OCT, and histopathological diagnoses (*OLP—Oral Lichen Planus; PV—Pemphigus Vulgaris; MMP—Mucous Membrane Pemphigoid*).

DGs Diseases	Clinical Hypotheses N (%)	OCT Diagnosis N (%)	Histopathological Diagnosis N (%)
**OLP**	**34/43 (79%)**	**33/43 (77%)**	**33/43 (77%)**
*Erosive OLP*	24/34 (71%)	22/33 (67%)	32/33 (97%)
*Bullous OLP*	10/34 (29%)	11/33 (33%)	1/33 (3%)
**PV**	**4/43 (9%)**	**4/43 (9%)**	**4/43(9%)**
**MMP**	**5/43 (12%)**	**6/43 (14%)**	**6/43 (14%)**

## Data Availability

Data available on request.

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
