# Peer review of "Diagnostic Concordance between Optical Coherence Tomography and Histological Investigations for Immune-Mediated Desquamative Gingivitis: Observational Study"

_ijerph, 2021, doi:10.3390/ijerph18179095_

Round 1

Reviewer 1 Report

This is an interesting study aimed to validate diagnostic concordance between specific OTC patterns for Desquamative Gingivitis and definitive histological diagnosis. While the study is interesting, the authors should address several points and enhance the discussion section.

L75-78: has any study investigated the feasibility of using OCT for DG diagnosis?

L 88: check the finishing date of patients’ recruitment.

L100: it would be useful to provide more details on the baseline questionnaire. For example, what sociodemographic variables were tested? And how medical history was obtained? self-report, medical records, use of medication?

It would be useful to add a table that presents the characteristics of the study participants.

Table 2: The OLP categories need alignment.

L 179: who made these hypotheses? Provide more information on the methods.

The first part of the discussion is very similar to the background. Discussion should start with the main findings of the study then comparing the findings with previous studies and the implications of the findings. Modify, please.

Author Response

Response to Reviewer 1 Comments

Review report:

This is an interesting study aimed to validate diagnostic concordance between specific OTC patterns for Desquamative Gingivitis and definitive histological diagnosis. While the study is interesting, the authors should address several points and enhance the discussion section.

Point 1. L75-78: has any study investigated the feasibility of using OCT for DG diagnosis?

Response 1. Thanks for the question. Our group has just published a systematic review on the subject (https://doi.org/10.3390/diagnostics11081453), the results of which have been appropriately introduced into the section.

Point 2. L 88: check the finishing date of patients’ recruitment.

 Response 2. It has been done.

Point 3. L100: it would be useful to provide more details on the baseline questionnaire. For example, what sociodemographic variables were tested? And how medical history was obtained? self-report, medical records, use of medication?

It would be useful to add a table that presents the characteristics of the study participants.

Response 3. Thanks for requesting details. This information has been added in the text (both in the M&M and in the Results sections), and a table with the resulting characteristics of the study participants has been inserted in the Results section.

Point 4. Table 2: The OLP categories need alignment.

Response 4. It has been done.

Point 5. L 179: who made these hypotheses? Provide more information on the methods.

Response 5. Thanks for requesting details. Appropriate more information has been added in the M&M section.

Point 6. The first part of the discussion is very similar to the background. Discussion should start with the main findings of the study then comparing the findings with previous studies and the implications of the findings. Modify, please.

Response 6. Thank you for your suggestion. The paragraph has been heavily skimmed, and we have left only some parts that can better introduce our results.

Finally, a complete minor revision of the English, as requested, was performed.

Reviewer 2 Report

Thank you for inviting me to review the manuscript entitled “Diagnostic concordance between Optical Coherence Tomography and histological investigations for immune-mediated Desquamative Gingivitis: observational study”. In this manuscript, the authors investigated the diagnostic concordance of specific Optical Coherence Tomography (OTC) in oral Lichen Planus (OLP), Pemphigus Vulgaris (PV), and Mucous Membrane Pemphigoid (MMP).

In this manuscript, the authors found that the diagnostic pattern of OTC in 34 patients was almost consistent with histological diagnosis. The possibility of using OTC in early diagnosis of patients with such immunological disease is suggested.

This is an interesting manuscript about the availability of OTC in early diagnosis in case of some immunological disease in oral cavity. The authors provided well-conducted statistical analysis. The study is well designed, and the material and methods are appropriate. References are up to date and acceptable; the text is well written. I do not have major critical comments.

Author Response

Response to Reviewer 2 Comments

Review report:

Thank you for inviting me to review the manuscript entitled “Diagnostic concordance between Optical Coherence Tomography and histological investigations for immune-mediated Desquamative Gingivitis: observational study”. In this manuscript, the authors investigated the diagnostic concordance of specific Optical Coherence Tomography (OTC) in oral Lichen Planus (OLP), Pemphigus Vulgaris (PV), and Mucous Membrane Pemphigoid (MMP).

In this manuscript, the authors found that the diagnostic pattern of OTC in 34 patients was almost consistent with histological diagnosis. The possibility of using OTC in early diagnosis of patients with such immunological disease is suggested.

This is an interesting manuscript about the availability of OTC in early diagnosis in case of some immunological disease in oral cavity. The authors provided well-conducted statistical analysis. The study is well designed, and the material and methods are appropriate. References are up to date and acceptable; the text is well written. I do not have major critical comments.

Response:

We would like to thank the reviewer for his detailed and favorable report. We are very pleased to have met his standards of adequacy and quality.

A complete minor revision of the English, as requested, was performed.

Reviewer 3 Report

The manuscript entitled “Diagnostic concordance between Optical Coherence Tomography and histological investigations for immune-mediated Desquamative Gingivitis: observational study” aimed to evaluate diagnostic concordance between specific Optical Coherence Tomography (OTC) patterns for DG related to oral Lichen Planus, Pemphigus Vulgaris, and Mucous Membrane Pemphigoid, and definitive histological diagnosis. The study is original and could provide advances in current knowledge. I have few comments that the authors may take into consideration.

Line  24 – Please change “present/absent of acantholytic cells” into “presence/absence of acantholytic cells”.

Lines 33-34 – Please use MeSH keywords.

Lines 169-17 – Please describe how the estimation of the number of lesions needed to validate the results, was done.

Lines 175-176 – What exactly does “unselected manner” refer to? The paragraph would need some more clarifications.

Lines 181-182 – Please add percentages in all brackets.

Lines 201-202 – The specific diagnostic isn’t clearly described. How exactly did the OCT method identify bullos OLP lesions in comparison to the histological method? What is the gold standard? Also figure 3 doesn’t give enough information about the specificity of OCT.

Lines 273-275 – This paragraph isn’t clear. Please rephrase.

Line 365 – Please change “zbased on OCT”.

Author Response

Response to Reviewer 3 Comments

Review report:

The manuscript entitled “Diagnostic concordance between Optical Coherence Tomography and histological investigations for immune-mediated Desquamative Gingivitis: observational study” aimed to evaluate diagnostic concordance between specific Optical Coherence Tomography (OTC) patterns for DG related to oral Lichen Planus, Pemphigus Vulgaris, and Mucous Membrane Pemphigoid, and definitive histological diagnosis. The study is original and could provide advances in current knowledge. I have few comments that the authors may take into consideration.

Point 1. Line 24 – Please change “present/absent of acantholytic cells” into “presence/absence of acantholytic cells”.

Response 1. It has been done.

Point 2. Lines 33-34 – Please use MeSH keywords.

Response 2. It has been done.

Point 3. Lines 169-17 – Please describe how the estimation of the number of lesions needed to validate the results, was done.

Response 3. Thanks for requesting details. Appropriate more information on sample size calculation has been added in the M&M section.

Point 4. Lines 175-176 – What exactly does “unselected manner” refer to? The paragraph would need some more clarifications.

Response 4. The sentence was deleted because it was confusing. Patients were progressively selected according to the inclusion criteria already detailed in the M&M section.

Point 5. Lines 181-182 – Please add percentages in all brackets.

Response 5. It has been done.

Point 6. Lines 201-202 – The specific diagnostic isn’t clearly described. How exactly did the OCT method identify bullos OLP lesions in comparison to the histological method? What is the gold standard? Also figure 3 doesn’t give enough information about the specificity of OCT.

Response 6. Thanks for the request for clarification.

In figure 3 we show OCT and histological comparison models for DGs, by erosive and bullous OLP, PV MMP, recovering the best images for the visualization of the parameters considered most discriminative. (i.e., morphology and localization of blisters, status of the basal membrane, epithelial thickness, presence/absence of acantholytic cells into blister and/or inflammatory infiltrate). Regarding bullous OLP, we recovered the images (OCT and histological, Figure 3, c/d) of the only 1 case diagnosed as bullous OLP, both by OCT and by histological examination after biopsy (considered the gold standard), just to show the diagnostic concordance of the patterns selected between the two techniques.

We would like to underline that this was the main objective of our study. Given the very encouraging results of this preliminary investigation, we hope to carry out other diagnostic studies in the near future, on a larger sample size, in order to validate their specificity.

Point 7. Lines 273-275 – This paragraph isn’t clear. Please rephrase.

Response 7. We tried to make it clearer.

Point 8. Line 365 – Please change “zbased on OCT”.

Response 8. It has been done.

Finally, a complete minor revision of the English, as requested, was performed.

Round 2

Reviewer 3 Report

Thank you for the clarifications.